# A Review on Potential Biofuel Yields from Cover Crops

**Liangcheng Yang** [1,2,*] , **Lucas D. Lamont** [1] , **Shan Liu** [3] , **Chunchun Guo** [3] and **Shelby Stoner** [2]

1    Department of Health Sciences Environmental Health and Sustainability Program, Illinois State University, Normal, IL 61790, USA
2    Department of Agriculture, Illinois State University, Normal, IL 61790, USA
3    College of Engineering, China Agricultural University, Beijing 100083, China
*    Correspondence: lyang@ilstu.edu; Tel.: +1-(309)-438-7133

**Abstract:** Millions of hectares of cover crops are planted in the U.S. and European Union to manage soil erosion, soil fertility, water quality, weeds, and climate change. Although only a small percentage of cover crops are harvested, the growing cover crop planting area provides a new biomass source to the biofuel industry to produce bioenergy. Oilseed crops such as rapeseed, sunflower, and soybean are commodities and have been used to produce biodiesel and sustainable aviation fuel (SAF). Other cover crops such as cereal rye, clover, and alfalfa, have been tested on small or pilot scales to produce cellulosic ethanol, biogas, syngas, bio-oil, and SAF. Given the various biofuel products and pathways, this review aimed to provide a comprehensive comparison of biofuel yield from different cover crops and an overview of the technologies that have been employed to improve biofuel yield. It was envisioned that gene-editing tools might be revolutionary to the biofuel industry, the work on cover crop supply chain will be critical for system scaleup, and high-tolerant technologies likely will be needed to handle the high compositional heterogeneity and variability of cover crop biomass for biofuel.

**Keywords:** cover crop; biofuel; sustainable aviation fuel; gene-editing; pennycress

## 1. Introduction

The U.S. Department of Agriculture (USDA) defines cover crops as grasses, legumes, and forbs for seasonal cover and other conservation purposes, primarily used for erosion control, soil health improvement, and water quality improvement [1]. Cover crops also mitigate greenhouse gas (GHG) emissions by increasing soil organic carbon (SOC) and reducing soil $N_2O$ and $CH_4$ emissions [2,3]. It was shown that the total amount of plant C added to soil with cover crop translated into greater SOC content by 10–20 Mg C/ha than the no-cover crop control [4]. Therefore, in recent years, cover crops have been widely adopted as a conservation practice in the U.S. and European countries. Commonly planted cover crops include grasses (i.e., ryegrass), legumes (i.e., peas and clover), brassicas (i.e., radishes), and non-legume broadleaves (i.e., spinach and flax). Rye and winter wheat are the two most common cover crops in the U.S. According to a report released in 2021, about one-third of the land planted with a cover crop received a financial assistance payment from either Federal, State, or other programs that support cover crop in 2018 [1]. In 2023, the USDA-ARS (Agricultural Research Service) released the Cover Crop Chart V 4.0 [5], which provides decision aids in selecting and managing 70 cover crop species that may be planted individually or in mixtures.

U.S. and European farmers are rapidly expanding the adoption of cover crops. In the U.S., farmers reported planting 6.23 million hectares of cover crops in 2017, a 50% increase compared to the 4.17 million hectares reported in 2012 [1]. Considering the USDA Census of Agriculture is conducted every five years, researchers have estimated that the U.S. cover crop adoption was roughly 8.09 million hectares in 2020 and has been projected to reach 40.47 million hectares by 2025 [6]. In 2017, East Coast states had the highest adoption

rates, i.e., Maryland (29%), Delaware (20%), Connecticut (15%), and New Jersey (14%). The adoption rates in most Midwestern states were 2–5% by 2017 but has reached 7.2% in 2021 [7]. In the European Union, cover crops are considered a viable option to mitigate GHG emissions and the adoption rate has increased from 6.5% of all agricultural lands in 2010 to 8.9% in 2016 and likely will continue to increase in the future [8]. Although not widely adopted yet, cover crops have been tested in many other countries, such as China [9], Japan [10], Thailand [11], and South Africa [12].

Currently, most cover crops are terminated with herbicide, tillage, or harvested for hay or silage to prepare for the planting of cash crops. Farmer may terminate their cover crop by grazing livestock on the forage. Cover crops may also be left on the field, be used to build up the soil humus layer so that increases soil fertility. The 2021 USDA-ARS report shows that harvesting is a common termination method for cover crops in soybeans (~17%) and corn-for-silage (~26%) [1]. Also, in about 10% of the soybean fields in 2018, farmers self-reported planting cover crops and harvesting that cover crop for grain, which is considered as double cropping according to the USDA practice standard.

The growing cover crop planting area provides a new biomass source to the biofuel industry. Soybean, rapeseed, sunflower, and peanut are the four major oilseed crops, with an annual production of 369.74, 88.54, 52.44, and 49.31 million tonnes in 2022–2023 [13]. The 2020 variety tests conducted in Tennessee reported mean averages of 2.25, 3.58, and 3.36 tonnes per hectare of dry matter biomass yields for brassicas, cereals, and legumes, respectively [14]. It was estimated that 109–154 million tonnes of biomass could be harvested if cereal rye was grown as a cover crop after corn and soybean harvest in the U.S. This amount of biomass has a liquid fuel potential comparable to the current U.S. ethanol industry [15]. Harvesting cover crop and using it for biofuel production is promising as the U.S. is taking large steps to move towards a low-carbon bioeconomy. In December, 2021, President Biden signed an executive order to reduce $CO_2$ emissions across federal operations by 65% by 2030 and be net-zero emissions by 2050 [16]. Carbon intensity (CI) scores are used to compare the amount of $CO_2$ released from various fuels. CI scores measure the amount of carbon dioxide equivalent ($CO_{2e}$), by weight, emitted per unit of energy consumed for a given fuel (e.g., g $CO_{2e}$/MJ) during its life cycle. Converting cover crops for fuel and energy production has a great potential to reduce CI scores. For example, the well-to-wheels emissions of carinata and rapeseed (canola) oils to biodiesel pathways were estimated to be 26.1 and 30.5 g $CO_{2e}$/MJ, much lower than the 90 g $CO_{2e}$/MJ from petroleum diesel [17].

To date, many studies have been conducted to produce a variety of biofuels, including biodiesel, ethanol, biogas, syngas, bio-oil, and sustainable aviation fuel (SAF), from cover crops. Except for the biodiesel production from oilseed crops, many studies were carried out on small or pilot scales, and the results varied due to different operation conditions and locations. This review aims to comprehensively summarize these studies to provide meaningful comparisons and an overview of the status of cover crop to biofuel technologies. Particularly, we will focus on the energy yield from cover crops and the methods used to improve energy production. Economic feasibility and environmental and social impacts, although important, are not the focus of this study.

## 2. Methods

The literature search for this review paper was conducted in 2023. There are many different cover crops being tested by researchers or planted by farmers. We focused on the ones listed on the USDA Cover Crop Chart V4.0 [5] and pennycress, which is an extensively researched oilseed cover crop. We prioritized field and bench test results and reports after 2019. There are numerous reports on oil content and yield from oilseed cover crops. We selected up to three reports for each oilseed cover crop. In total, we screened over 600 pieces of literature and discarded irrelevant ones for further analysis. A total of 119 pieces of literature were cited in the review.

Different units were found in the literature. For comparison purposes, we converted the results from the literature to the same SI units. For example, we use L-CH$_4$/kg-VS for biomethane yield and kg/ha for oil yield, respectively.

## 3. Results and Discussion

### 3.1. Cover Crop for Biodiesel Production

Biodiesel is a renewable and biodegradable fuel that can be manufactured from vegetable oils, animal fats, or recycled restaurant grease. Biodiesel has a heating value of 39–41 MJ/kg. It has attracted attention as a potential alternative to petroleum diesel. In 2022, the total volume of biodiesel production in the U.S. amounted to approximately 1.6 billion gallons [18]. According to the USDA Economic Research Service, 0.81 billion gallons of biodiesel was produced from soybean oil in 2022 [19]. Besides soybeans, extensive studies have been conducted to extract oil from other oilseed crops and use that oil for biodiesel production. The oil content and yield from various oilseed cover crops are summarized in Table 1. Their oil content and yield are affected by many factors, including the specie/variety, climate/region, soil conditions, and agricultural management, such as seeding rate, fertilizer application, and irrigation. The difference in oil yield can be significant. For example, field experiments conducted at five locations in western Canada in 2008–2009 showed the lowest and highest soybean oil yields were 70 kg/ha and 470 kg/ha, respectively. Therefore, results from up to three experiments at different cultivation conditions are included in Table 1 to provide a comprehensive comparison of oil content and yield. The location, soil and climate conditions, and years were provided as well. Many cover crops have comparable soil yield as soybean with higher oil content, therefore, have the potential to be used as feedstock in commercial scale biodiesel plants.

**Table 1.** Oil content and yield from cover crops and soybean.

| Cover Crop | Oil Content (%) | Oil Yield (kg/ha) | Location/Condition | Year | Ref. |
|---|---|---|---|---|---|
| *B. Carinata* | 33.2–36.0 | 388–400 | Seven western states in the US | 2013–2016 | [20] |
| *B. Carinata* | 38.0 | 380–920 [1] | Five western states in Canada | 2008–2009 | [21] |
| *B. Carinata* | 45.9–47.1 46.2–46.6 | 678–903 [2] 601–835 [2] | Alabama and Florida, US/Luverne, Dothan, Red Bay fine sandy loam | 2017–2019 | [22] |
| Camelina | 36.0 | 408 | Seven western states in the US | 2013–2016 | [20] |
| Camelina | 36.9–41.4 | 375 | Forest-Steppe, Steppe zones in Ukraine | 2001–2019 | [23] |
| Camelina | 40 (max.) | 740 (max.) | Arizona, US/Casa Grande sandy loam | 2013–2014 | [24] |
| Rapeseed (Canola) | 40.5–42.5 | 793–978 | Texas, US/arid, saneli silty clay loam | 2016–2017 | [25] |
| Rapeseed (Canola) | n.a. | 314–773 | New Mexico, US/semiarid, olton clay loam | 2015–2016 | [26] |
| Rapeseed (Canola) | 39.8–43.5 | 388–630 | Seven western states in the US | 2013–2016 | [20] |
| Brown mustard [3] | 37.7–39.3 | 447–473 | Seven western states in the US | 2013–2016 | [20] |
| Brown mustard | 35.8 | n.a. | North Dakota, US | n.a. | [27] |
| Brown mustard | 42.0 | 640–1290 [1] | Five western states in Canada | 2008–2009 | [21] |
| White mustard | 24.1–24.5 | n.a. | North Dakota, US | n.a. | [27] |
| White mustard | 26.0–26.6 | 253–295 | Seven western states in the US | 2013–2016 | [20] |
| White mustard | 30.0 | 380–650 [1] | Five western states in Canada | 2008–2009 | [21] |
| Field mustard [4] | 39.1 | 354 | Seven western states in the US | 2013–2016 | [20] |
| Field mustard | 45.0 | 450–1080 [1] | Five western states in Canada | 2008–2009 | [21] |
| Flax [5] | 37.1–42.8 | n.a. | Texas US/Ships clay and San Saba clay | 2008–2011 | [28] |

**Table 1.** *Cont.*

| Cover Crop | Oil Content (%) | Oil Yield (kg/ha) | Location/Condition | Year | Ref. |
|---|---|---|---|---|---|
| Flax | 36.9–40.3 | 633–827 | Northwest China/Ustorthents soil | 2012–2015 | [29] |
| Flax | n.a. | 644–845 | Argentina/Argiudol aquic soil | 2008 | [30] |
| Pennycress | 30–35 | 754 [6] | Midwest US | n.a. | [31] |
| Pennycress | 32.7<br>34.2, 33.5<br>30.6 | n.a. | Allartos, Greece/sandy loam<br>Bologna, Italy/silty clay loam<br>Illinois, US/silty clay loam | 2013–2015 | [32] |
| Pennycress | 34.5–36.7 | n.a. | Spain/semiarid Mediterranean | 2016–2017 | [33] |
| Peanut | 53.9 | 958 [7] | Manamadurai, India | 2012 | [34] |
| Peanut | 46.8–50.8 | n.a. | Uzbekistan/gray soil | 2012–2014 | [35] |
| Peanut | 45–52 | 1178 [8] | Average conditions in the US | n.a. | [36] |
| Sunflower | n.a. | 304–660 [9] | Average conditions in the Tanzania | 2010–2019 | [37] |
| Sunflower | 40–43 | 529–663 | Sergipe, Brazil/semiarid | 2018 | [38] |
| Sunflower | 49 (H31 hybrid) | 1504 | Tuscany, Italy/Mediterranean, clay loam | 2010 | [39] |
| Soybean | 19.5 | 568 [10] | Average conditions in the US | 2022 | [40] |

[1] The highest and the lowest data were excluded in the range to be more representative. [2] Converted from L/ha to kg/ha based on the density of 0.878 kg/L. [3] *Brassica juncea* L. also called Indian mustard, Chinese mustard, or oriental mustard. [4] *Brassica rapa*, also called turnip mustard, wild mustard, rape, or bird's rape. [5] *Linum usitatissimum* L., also known as common flax or linseed. [6] Converted from L/ha to kg/ha based on the density of 0.898 kg/L. [7] Converted from L/ha to kg/ha based on the density of 0.913 kg/L. [8] Converted from lb/ac to kg/ha. [9] Calculated based on 40% oil content. [10] Calculated based on 2021 soybean quality test results.

Field Pennycress (*Thlaspi arvense* L.) grows widely throughout temperate regions of the world and is extensively researched as it could serve as a winter oilseed cover crop. A winter oilseed crop would provide farmers the economic benefits of a new rotational crop transforming corn-soybean rotation into three cash crops. The oil yield was estimated to be 840 L/ha (754 kg/ha) in the U.S. Midwest Corn Belt [31]. However, in Europe, the reported seed yield (500–1400 kg/ha) was much lower than that in the U.S. (1100–2250 kg/ha), with a similar oil content of 30−37% [32]. In August, 2022, the multinational company, Bayer, acquired a majority share of CoverCress Inc. (Olivette, MO, USA), aiming to commercialize CoverCress^TM, a cover crop that was transformed from field pennycress, to produce low carbon biofuels [41]. A commercial launch was planned for 2022–2023, with the goal of growing to millions of acres across the lower US Midwest [42].

Peanut seed has a higher oil content (45–53.9%) than many other oilseed crops but the oil is commonly used as cooking oil for a better economic return, although it can be used for biodiesel production as well. Similarly, rapeseed (canola) and sunflower oil are commonly used as cooking oil. The competition between food and biofuel is often driven by market demand, policy, and climate change. Used cooking oil can be recycled and converted to biodiesel.

Converting plant oil to biodiesel has a high conversion rate. Depending on the oil composition and quality, a 90% conversion rate should be achievable for oil from most oilseed crops. It is common to have ~98% conversion rate from soybean oil to biodiesel with catalysts. It was reported that a conversion rate of 99.55% was achieved from rapeseed oil [43]. Some recent advances in conversion technologies are summarized in Table 2.

Various methods have been developed to improve biodiesel yield from cover crops. As shown in Table 2, these methods can be categorized into (1) improve cover crop to produce more oil or higher quality oil; (2) optimize agronomic practices to increase seed yield and oil yield; (3) increase oil extraction efficiency from seed; and (4) maximize the conversion of extracted seed oil to biodiesel. Efforts have also been made to improve the

efficiency of other steps within the cover crop-to-biodiesel process, including harvesting, cleaning, drying, storing, and pretreatment.

**Table 2.** Methods used to increase cover crop seed oil or biodiesel yield.

| | Methods | Description of Methods and Results | Ref. |
|---|---|---|---|
| Plant improvements | Breeding and gene editing | Gene editing tools such as CRISPR-Cas9 to develop traits with low seed coat fiber content and low erucic acid seed oil. | [44–46] |
| Agronomic management | Soil fertility and health | Apply fertilizer and/amendment to improve soil fertility & health. | [47,48] |
| | Irrigation and drainage | Manage irrigation and drainage to maintain soil moisture and nutrients and control groundwater levels. | [49,50] |
| Oil extraction | Mechanical extraction | Press at a pressure of 30–40 bar and temperatures of ~95 °C. | [51] |
| | Solvent oil extraction | Organic solvents i.e., n-hexane to speed up oil diffusion process. | |
| | Enzymatic oil extraction | Enzymes such as hemicellulases, cellulases to hydrolyze cell wall. | |
| | Supercritical fluid extraction | High-efficiency oil extract oil using solvents such as $CO_2$ at a supercritical condition. | |
| Conversion | Conventional esterification | Esterification or transesterification in the presence of alcohol. It can be acids, alkali, or enzyme-catalyzed. | [52] |
| | Supercritical transesterification | Transesterification under a supercritical condition with no or reduced amount of catalyst. | |
| | Superheated conversion | Reaction at a high temperature and is rapid. | |

Gene editing tools have been used to improve cover crop oil yield and showed promising results. For example, Tyler et al. cloned DGAT1 genes from other plants and over-expressed them in rapeseed (canola). In their greenhouse studies, the TmDGAT1 T1 transgenic lines exhibited oil content increases on a mature seed mass basis, ranging from 3.5–4.5%, representing net overall oil increases of 11–15% [44]. Phippen et al., estimated that 2465 kg ha$^{-1}$ of seed can be attained commercially from pennycress in the near term through breeding, gene editing, and improvements to agronomic practices. This represents an approximate 10% increase from wild pennycress [45]. The Clustered Regularly Interspaced Short Palindromic Repeats/CRISPR-associated proteins (CRISPR-Cas) have emerged as a breakthrough breeding tool in recent years. Studies have shown that this tool can be used to reduce plant biomass lignin content [53] and increase seed oil and fatty acid content in oilseed crops [54]. Although still under development, this revolutionary tool likely will make big differences to cover crop farming and biofuel industry.

Agronomic management, such as fertilizer and pesticide application, planting, tillage, and irrigation, are commonly used to improve cover crop yields. A recent study in India showed that sesame, pearl millet, and mustard yield increased 24.24%, 4.2%, and 8.4%, respectively, by applying nano-fertilizers of nitrogen and zinc along with the organic farming practices [48]. Similarly, by controlling drainage, soil moisture, and nutrients, Dou et al. increased oilseed sunflower yield by 4.52–11.14% and increased water-use efficiency by 1.16–10.8% [49]. The Midwest Cover Crops Council (MCCC) recommends agronomic management practices for different cover crops [55].

Oil extraction from seed meal and conversion to biodiesel have high conversion rates. A residual oil content of less than 0.5% for soybean meal is generally expected for today's soybean oil extraction plant, and a high degree (98%) of esterification can also be achieved. Most studies focus on energy efficiency, conversion rate, and processing speed. Among them, the use of catalysts and solvents, and pretreatment of oil have been heavily researched, while the use of supercritical fluids is a relatively new development. A supercritical fluid is any substance at a temperature and pressure above its critical point, where distinct liquid and gas phases do not exist. Supercritical fluid extraction (SFE) is mostly carried out using $CO_2$, which has low critical temperature and pressure. In addition

to being non-flammable and non-toxic, highly pure $CO_2$ is obtainable at low costs, and its complete removal from the extract can be accomplished with ease. SFE is known for reduced process time and high productivity. A review of using supercritical fluids for oil extraction has been published previously [56]. Similarly, oil conversion to biodiesel can be carried out using supercritical fluids, such as methanol, ethanol, and tert-butyl methyl ether (MTBE). Supercritical transesterification for biodiesel synthesis has been summarized by Karmakar and Halder [52]. It requires no, or just a small amount, of catalyst and can be completed within 10–40 min. For example, Farobie et al. achieved biodiesel yield as high as 93.8 wt% from rapeseed (canola) oil at 350 °C after 30 min, using supercritical 1-propanol [57].

### *3.2. Cover Crop for Ethanol Production*

The U.S. Renewable Fuel Standard (RFS) program required the production of 16 billion gallons of cellulosic biofuel by 2022 [58]. Although failed to achieve the goal, this program triggered extensive research on ethanol production from cellulosic biomass, including cover crops. Like corn stover, cover crop biomass (stalk, leave, stem, not including seed meal) contains high percentages of cellulose and hemicellulose, which can be hydrolyzed and then converted to glucose and xylose, respectively. The sugars then can be fermented to ethanol. Table 3 lists the potential ethanol yields from various cover crops. Most reported results are in the range of 0.1–0.3 g of ethanol per gram of dry matter. These results are comparable to corn stover ethanol yields, which were reported to be in the range of 0.17–0.20 g/g-dry matter by Tumbalam et al. [59], and 0.27 g/g-dry matter (after hydrothermal pretreatment) by Saha et al. [60], respectively. Also, as a comparison, the theoretical ethanol yield from glucose is 0.51 g/g.

**Table 3.** Reported ethanol yields from cover crop biomass.

| Cover Crop | Yield, g/g-Dry Matter | Ref. |
| --- | --- | --- |
| Faba bean | 0.083 g | [61] |
| Oilseed rape | 0.099 | [61] |
| Rapeseed straw | 0.11 | [62] |
| Rapeseed straw | 0.12–0.15 | [63] |
| Winter rye | 0.14 | [61] |
| Oat straw | 0.15 | [64] |
| White clover | 0.19 | [65] |
| Sweet sorghum stem | 0.22 | [66] |
| Sweet sorghum bagasse | 0.25–0.28 [1] | [67] |
| Proso millet | 0.28 | [68] |
| Triticale | 0.29 | [69] |
| Pearl millet | 0.30–0.33 [2] | [70] |
| Triticale | 0.33–0.34 | [71] |

[1] Estimated based on the published figure. [2] Calculated based on the ethanol density of 0.789 kg/L. Note: the heating value of ethanol is around 27 MJ/kg.

Due to the heterogeneous complexity and recalcitrance of lignocellulosic cover crop biomass, pretreatment is usually required to break the lignin seal and/or disrupt the structure of crystalline cellulose to increase the accessibility of cellulase enzymes, with the goal of increasing sugar and ethanol yields. Saccharification and fermentation (SSF) could be carried out simultaneously or separately. A few pretreatment methods have been tested, and their results are shown in Table 4. Acid and alkaline pretreatment can be carried out under relatively mild temperature and pressure conditions, therefore, can be more cost-effective than the steam explosion method. Other methods, such as enzymatic and hot

water pretreatment, could be used as well. Although not directly working on cover crops, gene-editing research has been conducted in recent years to improve ethanol production from lignocellulosic biomass. For example, Varize et al. overexpressed the MSN2 gene for ethanol fermentation from sugarcane molasses. As a result of the overexpression, the strain tolerance to ethanol was increased, which improves the fermentation performance under a high sugar content, and acid treatment between cycles [72].

**Table 4.** Technologies used to increase ethanol yield from cover crops. DM: dry matter.

| Methods | Ethanol Yield Increase | Ref. |
|---|---|---|
| Alkaline pretreatment | Triticale: from 0.17 to 0.29 g/g DM | [69] |
| Alkaline pretreatment | Rye straw: increased to 0.081 g/g DM | [73] |
| Acid pretreatment | Wheat: from $13 \pm 2$ to $17 \pm 0$ g/L | [60] |
| Acid pretreatment | Rye straw: increased to 0.096 g/g DM | [73] |
| Wet oxidation pretreatment | Winter rye: from 0.0059 to 0.14 g/g DM<br>Oilseed rape: from 0.0097 to 0.099 g/g DM<br>Faba bean: from 0.010 to 0.083 g/g DM | [61] |
| Steam explosion | Oat: from 0.085 to 0.15 g/g DM | [64] |

### 3.3. Cover Crop for Biogas Production

Anaerobic digestion (AD) is a process through which bacteria break down organic matter into biogas in the absence of oxygen. AD is known as a highly tolerant biorefining technology and has been used to produce biogas from diverse organic wastes with varying compositions [74–77]. The biogas from AD generally contains 60–70% methane ($CH_4$) and 30–40% $CO_2$ and can be used to produce energy in the form of heat, electricity, and transportation fuels [78,79]. Extensive research exists on AD of lignocellulosic biomass such as corn stover, switchgrass, rice straw, and giant reed [80–83]. However, there are only a few studies on AD of cover crop biomass, mainly cereal rye. The results are included in Table 5 and are comparable to corn stover biomethane yield, which is typically in the range of 250–350 L-$CH_4$/kg-VS (volatile solids), depending on the pretreatment and operation conditions [84]. Pretreatment and continuous stirring improved the biomethane yields.

**Table 5.** Biomethane yields from cover crops.

| Cover Crop | Yield, L-$CH_4$/kg-VS | Conditions | Ref. |
|---|---|---|---|
| Cereal rye | 99–347 | Sulfuric acid pretreated | [85] |
| Cereal rye | 175–225 | Size reduced | [86] |
| Cereal rye | 348 | Alkaline pretreated | [87] |
| Cereal rye | 360 | 42 °C, continuous stirring | [61] |
| Radish | 150–210 | Co-digestion | [88] |
| Barley straw | 283 | Co-ensiling | [89] |
| Oat | 294 | Alkaline pretreated | [87] |
| Red clover | 319 | Co-ensiling | [89] |
| Oilseed rape straw | 420 | 42 °C, continuous stirring | [61] |
| Faba bean straw | 440 | 42 °C, continuous stirring | [61] |

Note: Methane has a heating value of 35.8–39.8 MJ/Nm$^3$.

Challenges in AD of cover crops include recalcitrance of lignin, biomass floating, varying C/N ratio, harvest and storage, and relatively low methane yield. Co-digestion, pretreatment, and ensiling are three methods that have been used to improve biogas yield and digester stability. Belle et al. co-digested radish cover crop and dairy manure, and

reported that methane production increased by 39% with radish addition compared to manure only [88]. Domanski et al. pretreated 10 g rye straw biomass with up to 10% 100 mL $H_2SO_4$ and increased methane yield from 107.21 to 347.42 with one hour of treatment at 121 °C. However, this method may not be economically feasible due to the high acid over biomass ratio. Petersson et al. applied wet oxidation to pretreat cereal rye and oilseed rape straw and increased the methane yields by 34% and 8.5% respectively. Interestingly, their results showed that the wet oxidation pretreatment slightly decreased methane yield from Faba bean straw. They used small 100 mL serum flasks and only 0.3 g dry straw for the methane yield experiment, which could increase uncertainty in their results [61]. Vlierberghe et al. investigated the effects of concurrent storage and alkali pretreatment on AD of cover crops (cereal rye and sunflower) and reported no changes in biogas yield [87,90]. Feng et al. co-ensiled cover crops (red clover, Trifolium pretense) and barley straw without additives and used the silage for biogas production. Their results showed that co-ensiling improved stability and increased methane yield by approximately 5–10% [89]. Furthermore, they noted that manure addition to the digestion process led to a higher methane yield and buffer capacity [91].

Although there is great potential in cover crop AD for biomethane production, so far, only bench and small pilot scale studies have been conducted, and this presents a challenge in the evaluation of its economic feasibility. Yang et al. estimated the potential revenue of bioenergy produced from a commercial-scale cover crop AD facility but didn't assess the costs [86]. Igos et al. provided the first overview of the environmental and economic consequences of co-digesting rye and maize for energy production, with large uncertainties [92]. More research is needed in this area to evaluate the feasibility of a system scaleup.

### 3.4. Cover Crop for Syngas and Bio-Oil Production

Like other lignocellulosic biomass, cover crops can be used as feedstock for pyrolysis. Table 6 shows the non-condensable gas and bio-oil yields from some cover crop biomass and processing residuals using different types (flash, fast, and slow) of pyrolysis. The non-condensable gas contains $H_2$, CO, $CO_2$, and small molecular carbohydrates, therefore, can be purified to syngas. The distribution of products (gas, oil, and char) from pyrolysis is mainly determined by the feedstock composition (hemicellulose, cellulose, lignin, and moisture) and the reaction conditions. Boateng et al. obtained a higher non-condensable gas yield but a lower bio-oil yield from the early bud stage alfalfa stems than the full-flower stage. This is likely due to the difference in cellulose and lignin content between the two maturity stages [93]. In another study, they also showed that the yields of non-condensable gas and bio-oil increased with pyrolysis temperature for alfalfa and reed canarygrass (an invasive cover crop) [94]. In addition to pyrolysis time and temperature, microwave, and catalysts have been used to improve the pyrolysis performance. Those studies have been previously reviewed and summarized [95,96].

The non-condensable gas from lignocellulosic biomass pyrolysis could contain 25–30% and 23–33% of $H_2$ and CO, respectively [97], and could have a heating value of approximately 17 MJ/Nm$^3$. Typical bio-oil from biomass pyrolysis could have a heating value of 16–19 MJ/kg, which is 40–50% of diesel fuel [98]. Both non-condensable gas and bio-oil have the potential to be refined to produce value-added biofuels or bioproducts.

**Table 6.** Non-condensable gas and bio-oil yields from cover crop biomass and processing residual.

| Feedstock | Conditions | Non-Condensable Gas Yield, wt% of Dry Biomass | Bio-Oil Yield, wt% of Dry Biomass | Ref. |
|---|---|---|---|---|
| Alfalfa stems (early bud) | Fluidized-bed, fast pyrolysis | 16.3 | 45 | [93] |
| Alfalfa stems (full flower) | | 12.8 | 53 | [93] |
| Fodder radish seed cake | Slow pyrolysis | 16.2 | 57.8 | [99] |
| Rapeseed seed cake | Fixed bed reactor | 8–14 | 48–62 | [100] |
| Rapeseed seed cake | Flash pyrolysis | 24 | 47 | [101] |

Note: Non-condensable gas heating value is around 17 MJ/Nm$^3$. The pyrolysis bio-oil heating value is 16–19 MJ/kg.

### 3.5. Cover Crop for SAF Production

Aviation fuels are used in gas-turbine powered aircraft. The main components are linear and branched alkanes and cycloalkanes with a typical carbon chain length distribution of C8 to C18, where the ideal carbon chain length is C8–C16. SAF has a lower heating value of around 42.8 MJ/kg. SAF can be produced from many sources, including waste fats, oils and greases, municipal solid waste, agricultural and forestry residues, wet wastes, and non-food crops such as cover crops. SAF reduces GHG emissions. The International Civil Aviation Organization (ICAO) evaluated life-cycle emission values of SAFs from various feedstock and conversion processes with the impact of induced land use change (ILUC). The CI of SAF derived from rapeseed oil (47.4 gCO$_{2e}$/MJ) is 35.9% lower than the CI of petroleum jet fuel (74 gCO$_{2e}$/MJ) [102]. The International Air Transport Association estimated that SAF could contribute around 65% of the reduction in emissions needed by aviation to reach net-zero in 2050 [103].

Producing SAF from cover crops is still at an early developmental stage. It was estimated that SAF yield could be 357 L/ha for pennycress [104] and 524 L per tonne of carinata seed [105], based on modeling results. So far, there are no pilot or full-scale studies on SAF production directly from cover crops. It has been reported that scientists with the USDA-ARS have developed a way to make jet biofuel from soybean oil [106]. The yield information is currently not available. In 2022, the U.S. EPA approved rapeseed oil for use as SAF feedstock under the RFS, allowing hydrotreated fuels made from rapeseed oils to qualify for the program and generate renewable identification number (RIN) credits [107]. Technologies for producing renewable aviation fuel by hydro-processing of biomass were reviewed by Why et al. [108]. Catalysts such as Ni-Mo/HY, Pt/Al$_2$O$_3$, and Pt/SAPO-1 have been tested in the conversion of oilseed cover crops (i.e., camelina, carinata, and rapeseed) to SAF, with a yield of 48–53.5% [108,109].

Technically, SAF can also be produced from biogas, landfill gas, syngas, or ethanol oriented from cover crops. In February 2021, NetJets, an American private jet company, announced it to invest in WasteFuel, a Los Angeles-based company that transforms landfill waste into SAF and purchase a minimum of 100 million gallons of WasteFuel's SAF over the next ten years [110]. In March 2022, the U.S. Department of Energy (DOE) funded SkyNRG Americas to conduct feasibility studies and small-scale research and development to produce SAF from biogas [111]. In May 2023, Kenya Airways became the first African airline to use SAF produced from waste animal fats and used vegetable oils by Eni [112]. Possible pathways for transforming cover crops to biogas, and then to SAF are shown in Figure 1. Cleaned biogas can be converted into syngas via steam reforming, dry reforming, or partial oxidative reforming [78]. Then, conditioned syngas can be transformed into aviation fuel via Fischer-Tropsch synthesis. Alternatively, methanol and alcohol can be produced from methane gas via a chemical or biological process [79] and then transformed to aviation fuels via the A-t-J (Alcohol-to-Jet fuel) process [113]. Syngas and alcohol may also be produced from cover crops via pyrolysis or fermentation respectively.

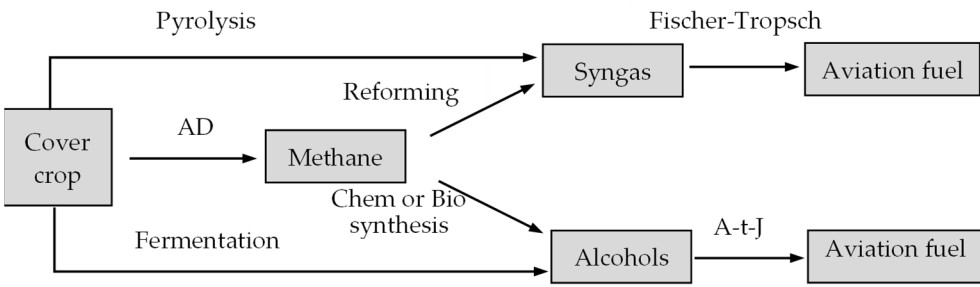

**Figure 1.** Possible pathways of converting cover crop to SAF. Developed based on [113].

### 4. Conclusions and Perspectives

There are two approaches to use cover crops for energy production: (1) use oil in oilseed crops such as rapeseed, sunflower, soybean, to produce biodiesel or SAF; and (2) use the plant biomass to produce cellulosic ethanol, biogas, syngas, bio-oil, and SAF. The technologies and market for biodiesel from oilseed crops have been well established, although there are still challenges such as the lack of modernization, low oil recoveries, lack of standardization of product quality among small-scale processors [114]. The demand for oilseed crops likely will increase, therefore, improving current technologies and infrastructure, and developing new oilseed crops are important. The work on winter oilseed crops, such as pennycress, have shown promising results. The commercial launch of the CoverCress$^{TM}$ (transformed from pennycress) likely will bring this cover crop one step closer to full commercialization. The recent development in gene-editing tools has the potential to improve winter oilseed crop oil yield and quality.

Using cover crop biomass for ethanol, biogas, syngas, bio-oil, and SAF productions has met with more challenges and the technologies and market are currently not well established. Producing ethanol and biogas from cover crop biomass has been studied at pilot scales. Producing SAF from cover crop biomass is at an early developmental stage. One challenge in cover crop bioconversion is the cover crops' compositional heterogeneity and variability. The heterogeneity and variability are caused by the diverse cover crop varieties, different harvest times, and storage methods. For example, cereal rye would be harvested before maturity in temperate region [115], leading to structural differences from fully senesced crops as its water-soluble fraction of dry matter decreases with increasing maturity substantially [15]. Cover crops accumulate more carbon than nitrogen as mature, leading to increase of carbon (C) to nitrogen (N) ratio [116]. Similarly, during the wet ensiling storage, lactic acid-producing bacteria ferment substrates within the feedstock into lactic acid, resulting in compositional change and a pH reduction [117,118]. The recalcitrance of lignin and low energy yield are two other challenges for cover crops like many other lignocellulosic biomass. The biofuel yield from cover crops varies significantly in the reported studies, due to the different specie/variety, climate/region, soil condition, agricultural management, and processing methods. The typical biofuel yields are 0.10–0.30 g, 0.45–0.55 g, and 0.10–0.15 g per g of cover crop dry matter for ethanol, bio-oil, and non-condensable gases, respectively, and 150–300 L/kg-VS for biomethane production. Methods have been tested to increase biofuel yield by: (1) breeding or gene editing cover crops to increase oil, reduce the lignin content, and improve biomass digestibility; (2) managing soil, water, and agronomic practices to increase seed and biomass yields; (3) improving conversion technologies to generate more energy from cover crops; and (4) optimizing supply chain (i.e., harvesting, drying, storing, transportation, and pretreatment) to preserve biomass matter and reduce costs. The gene-editing tools might be revolutionary to the industry, especially its applications in improving cover crop biomass digestibility and starch content. Optimization of the biomass supply chain is critical for system scale up, which requires work on modernization and standardization. The conversion efficiency improvement highly relies on technological breakthroughs, but also requires work on economic feasibility and environmental impacts. Many of the cover crop-to-energy pathways need to be tested

and optimized at a larger scale before commercialization. Given the compositional heterogeneity and variability of cover crop biomass, high-tolerant technologies and pathways such as AD, pyrolysis, and supercritical extraction likely will gain attention to be used as a starting technology or a platform prior to production of other higher value biofuels or bioproducts.

**Author Contributions:** Conceptualization, L.Y.; methodology, L.Y.; validation, L.Y.; formal analysis, L.Y.; investigation, L.Y., L.D.L. and S.S.; resources, L.Y. and S.L.; data curation, L.Y.; writing—original draft preparation, L.Y., L.D.L., C.G. and S.S.; writing—review and editing, L.Y. and S.S.; visualization, L.Y.; supervision, L.Y. and S.L.; project administration, L.Y.; funding acquisition, L.Y. All authors have read and agreed to the published version of the manuscript.

**Funding:** This study was funded by the USDA-NIFA NLGCA Capacity Building Grants for Non-Land Grant Colleges of Agriculture Program (Award number: 2020-70001-31279).

**Institutional Review Board Statement:** Not applicable.

**Informed Consent Statement:** Not applicable.

**Data Availability Statement:** All data are within the manuscript.

**Acknowledgments:** This study was supported by ISU Farm manager Jason Lindbom and his crews.

**Conflicts of Interest:** The authors declare no conflict of interest.

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
