# Peer review of "A Review on Potential Biofuel Yields from Cover Crops"

_fermentation, doi:10.3390/fermentation9100912_

Round 1

Reviewer 1 Report

This work provides important knowledge on the use of cover crops occurring in the U.S. and Europe as a potential raw material for the production of renewable energy. The article is generally legible, however, please see the following notes: - standardize the way of writing titles of tables and figure; - standardize the records of units, e.g. kg∙ha-1, not kg/ha; - Latin plant names should be in Italic; - lines 230 and 231 - remove line numbers from the table and the font should be black; - References - check and correct/complete entries, especially items 27, 41, 42, 48, 49, 56, 72, 82, 84, 94, 101, 104, 105.

Reviewer 2 Report

General comments:

Thank you for your submission, the review makes for a great read on the different biofuel yields of upcoming cover crops. The added value is limited as the report does not compare the different crops/technologies in terms of environmental impacts, feasibility or normalized process economics. The discussion/comparison of the technical readiness could also be expanded. Stronger recommendations should be made given the yields reported.

Specific comments:

Line 2: Mostly, the review summarizes biofuel yields, rather than comparing energy yields. An overview table/section on the latter would, however, be very interesting and, then (or if each table included energy columns), the title would be accurate.

Line 9: Provide at least the order of magnitude.

Line 10-11: Quantify harvested fraction.

Line 12: Careful with the term biorefining - typically it refers to multiple product/material streams, whereas the review discusses almost exclusively the sole production of individual biofuels.

Line 16: Energy vs biofuel yield as before.

Line 19: Biorefining or biofuel industry?

Line 77: Please substantiate this CI claim with examples.

Line 84: biorefining or biofuel production technologies?

Line 100: Kg/ha -> kg/ha

Line 122: The authors should specify why this seasonal opportunity could make additional sense.

Lines 129-132: The authors should highlight the ethics and trends behind competition with food and feed. E.g., through the lens of the water, food energy nexus.

Table 1:
"Ukrainian" - typo?
Pennycress 840 L/ha - express as kg/ha
d: Confusing for Europe, where canola is often called rapeseed oil
Poor geographical representation - missing crop yields from LatAm, Africa.

Line 240: Delete "worked"?

Line 269: The authors mention economic feasibility but so far only technological readiness is covered. TEA results could be presented for each technology, especially evaluating how the presented yield ranges correspond to feasibility, minimum sales prices to add merit to the review. Otherwise, the biofuel yields do not provide too much information for effective decision-making.

Section 3.4: The short section on pyrolysis seems to be the first that considers more than one product stream (biorefining). Others simply talk about biofuel production, the title and abstract should reflect this.

Section 3.5:
Eni produce SAF from vegetable oils grown in Africa. This case study could be included.
It should be made clearer that SAF production from cover crops is at very early developemental stages.

Line 301-302: Needs to be substantiated, covering caveats such as ILUC.

Lines 314-316: Is this also Why et al.? Needs clearer referencing.

Line 356: "biorefining" - Again, confusing use of the term. Lignin can be a beneficial value stream in a biorefinery but, correctly so, not so much when producing solely biofuel.

Line 367: Environmental impact of the different technologies was also not compared, limiting the added value of the review.
